# Model Development for Identifying Aromatic Herbs Using Object Detection Algorithm

**Samira Nascimento Antunes** [1], **Marcelo Tsuguio Okano** [1,*], **Irenilza de Alencar Nääs** [1], **William Aparecido Celestino Lopes** [1], **Fernanda Pereira Leite Aguiar** [1], **Oduvaldo Vendrametto** [1], **João Carlos Lopes Fernandes** [1] and **Marcelo Eloy Fernandes** [2]

1   Graduate Program in Production Engineering, Universidade Paulista, R. Dr. Bacelar 1212, São Paulo 04026-002, Brazil; samira.antunes@aluno.unip.br (S.N.A.); irenilza.naas@docente.unip.br (I.d.A.N.); william.lopes12@aluno.unip.br (W.A.C.L.); fernanda.aguiar11@aluno.unip.br (F.P.L.A.); oduvaldo.vendrametto@docente.unip.br (O.V.); joao.fernandes1@docente.unip.br (J.C.L.F.)

2   Faculdade de Tecnologia de Barueri, CEETEPS, Barueri 06401-136, Brazil; marcelo.fernandes3@fatec.sp.gov.br

\*   Correspondence: marcelo.okano1@docente.unip.br

**Abstract:** The rapid evolution of digital technology and the increasing integration of artificial intelligence in agriculture have paved the way for groundbreaking solutions in plant identification. This research pioneers the development and training of a deep learning model to identify three aromatic plants—rosemary, mint, and bay leaf—using advanced computer-aided detection within the You Only Look Once (YOLO) framework. Employing the Cross Industry Standard Process for Data Mining (CRISP-DM) methodology, the study meticulously covers data understanding, preparation, modeling, evaluation, and deployment phases. The dataset, consisting of images from diverse devices and annotated with bounding boxes, was instrumental in the training process. The model's performance was evaluated using the mean average precision at a 50% intersection over union (mAP50), a metric that combines precision and recall. The results demonstrated that the model achieved a precision of 0.7 or higher for each herb, though recall values indicated potential over-detection, suggesting the need for database expansion and methodological enhancements. This research underscores the innovative potential of deep learning in aromatic plant identification and addresses both the challenges and advantages of this technique. The findings significantly advance the integration of artificial intelligence in agriculture, promoting greater efficiency and accuracy in plant identification.

**Keywords:** aromatic herb; convolutional neural network; deep learning; computer vision; YOLO v8

## 1. Introduction

Since ancient times, aromatic herbs such as mint, basil, and rosemary have enhanced food flavor and aroma and prolonged shelf life due to their antiseptic properties [1]. These herbs play an integral role in daily life, with culinary and medicinal applications. Aromatic herbs are typically small plants whose leaves emit distinct aromas, occasionally leading to confusion with medicinal herbs. Although there are similarities, medicinal herbs have a broader range of uses involving various plant parts, whereas aromatic herbs are primarily valued for their leaves.

Aromatic plants have long been utilized in medicine, food preservation, seasoning, and religious ceremonies. The diversity among aromatic herbs is extensive, and their visual differentiation poses significant challenges. The enhanced visual characterization of these herbs can substantially benefit wholesalers, retailers, farmers, cooperatives, importers, exporters, and the agro-industry. This improved understanding facilitates accurately identifying desired herb types for purchase, sale, consumption, and registration purposes for research endeavors [2,3].

With the increasing prevalence of digital technologies and digital transformation, efforts have been made to observe how these technologies can help produce aromatic herbs.

In the digital transformation scenario, it is observed that new digital technologies facilitate improvements in business processes, simplify operations, and create new business models. Object detection and categorization in images is one of the most significant problems in computer vision and related fields [4]. Additionally, the process becomes more intricate due to the various perspectives, sizes, angles, perspectives, occlusions, and illumination.

Previous investigations [2,5,6] have attempted different approaches to improve the accuracy and efficiency of herb identification from images. However, there have been limitations regarding detecting the shape of herbs, a critical characteristic for identifying the leaf family and removing the background. These issues were investigated previously, but have not been solved.

One way to ensure better accuracy in image detection is with convolutional neural networks (CNNs), which, like a neural network in the human brain, is a circuit with connections of neurons with weights between nodes, where a positive weight represents a stimulating connection, while negative values represent inhibitory connections. With the development of technology, deep learning enables the development of an image object detection model. CNNs are amongst the most widely utilized deep learning models for image detection and classification, owing to their superior accuracy to other machine learning algorithms [7].

Owing to the substantial increase in the utilization of digital images, deep learning, an artificial intelligence (AI) method, through an image object detection scheme, can enhance customer experience by identifying diverse types of horticultural products in the supply chain, such as aromatic herbs. The research question is as follows: to what extent can a deep learning model employing the You Only Look Once (YOLO) v8 framework effectively differentiate and accurately identify various aromatic herbs, including rosemary, mint, and bay leaf?

The current study aimed to develop and present a procedure for identifying three types of aromatic herbs using object detection algorithms through YOLO v8 architecture. The present study is organized as follows: In addition to this introduction, Section 2 presents the background, Section 3 describes the methodology used, and Section 4 presents the results and summarizes the discussion. Finally, Section 5 presents the conclusion and future research.

## 2. Background

Object detection is a component of computer vision that involves recognizing and positioning objects in videos or images [8,9]. Object class detection is usually based on a set of features, meaning each object of a specific class has certain characteristics based on which it is classified into different classes. Deep learning is a machine learning method that allows machines to recognize patterns in data [9]. Given a large dataset of labeled examples, it enables the learning of the most predictive features directly from images. Within the machine learning scenario, CNNs have become extremely popular and effective [10]. A CNN comprises a series of layers, and each layer receives the previous layer's output as its input. The input plane receives images of characters with a standardized size and centeredness, and each layer's units receive input from a set of smaller units located in the previous layer [11]. Two unique attributes of the architecture of CNNs are their sparse connections and shared weights [12]. CNNs exploit spatially local correlation by imposing a pattern of local connectivity between neurons of adjacent layers, where units in layer m are connected to three spatially adjacent units in layer m − 1. Every convolutional filter in the layer is repeated across the entire layer, sharing the same weights and biases. Such an approach will help CNNs to generalize the identification of problems more effectively while also increasing the learning efficiency by reducing the number of free parameters to be learned.

Object detection is a fundamental and long-standing problem in computer vision. Its task is to generate bounding boxes (in 2D pixel coordinates) for detected objects in an image that belongs to pre-specified object classes and assign classification scores to them [13]. It has been observed that object detection has improved significantly due to deep

neural networks, with convolutional neural networks (CNNs) being the most prominent architectures. The window that slips between classifiers and single-shot convolutional neural networks generates similar high-density windows around a particular object's correct location. Non-maximum suppression (NMS) chooses the single most significant candidate within the cluster of detection results for each object in each image. Therefore, non-maximum suppression (NMS) is analogous to a traditional clustering problem and generally depends on two primary operations: determining the cluster assignment for each detection and identifying a representative for each cluster [14]. Sun et al. [11] explain how NMS eliminates redundant boxes and selects the candidate box that best represents the object. Meanwhile, the remaining candidate box is called the object representative. Among the various algorithms widely used in various object detector models, YOLO stands out.

Through a procedure called backpropagation, a CNN learns the weights and biases of the kernel from a collection of input images [15]. These values serve as parameters that encapsulate essential characteristics of the images, independent of their location. These kernel weights slide through an input image performing element-wise dot products, producing intermediate results that are subsequently summed with the learned bias value. Then, each neuron obtains an output based on the input image. These outputs are also called activation maps. To reduce the number of parameters and prevent overfitting, CNNs decrease the resolution of inputs using a different layer called pooling. Activation functions are employed in a CNN to augment the model with nonlinearity. This enables the model to learn more intricate patterns in data.

### 2.1. You Only Look Once (YOLO)

YOLO is an object detection approach that aims to predict which objects are present and where they are by looking at the image only once [16]. In recent years, there has been a rise in research articles addressing object identification using the YOLO algorithm. However, we found a few articles about using YOLO to detect herbs. There has been a theoretical basis for recognizing the rhizome and main root in plants through "internal content difference" and "external morphological difference" [17]. Weed detection was previously performed to assist laser weed removal robots [18]. Previous studies [3,19] determined YOLO's adaptability and efficiency in plant and herb identification through image analysis, leveraging its speed and accuracy in object detection tasks. The technology is beneficial for enhancing agricultural practices and scientific research by automating the extraction of detailed trait data from images.

Recent advances in research have demonstrated the effectiveness of image recognition systems based on YOLO. A notable example is the study by [20], which presents the implementation of an apple detection system using YOLO v3, and which was later integrated into a harvesting manipulator robot. The developed system improved apple identification accuracy under various lighting and obstruction conditions. Furthermore, the trajectory of the manipulator robot was optimized using reinforcement learning techniques, resulting in a 92% increase in accuracy in laboratory tests. Under controlled conditions, the system achieved a 100% recognition rate. These improvements concur with the increasing popularity of utilizing convolutional neural networks (CNNs) in agriculture, proving effective in solving complex computer vision tasks.

Similarly, a previous study was conducted [21] by applying YOLO v3 with advanced pre- and post-processing techniques for apple identification, highlighting the evolution of computer vision algorithms to meet the demands of modern horticulture. The improvements implemented resulted in reduced error rates and average identification time, making the system more efficient and comprehensive than previous versions, such as the faster region-based convolutional neural network (RCNN) and dynamic activation sparsity network (DaSNet) v2. This study highlights the importance of computer vision in agricultural automation.

On the other hand, the study by [22] discusses aspects related to object detection in images, covering everything from likelihood relationships to using scalable and efficient

neural networks. Given the increasing relevance of object identification and recognition tasks in images, the authors explore the development of convolutional neural networks to solve recognition problems. The study includes the evaluation of several pre-trained architectures, considering metrics such as error probabilities, precision, detection recall, intersection over union, and average interpolated precision.

Mirzaei et al. [23] contribute a comprehensive review on small object detection and tracking, addressing challenges and methods in computer vision. The study analyzes small object detection techniques, including methods based on CNNs, image processing algorithms, and machine learning techniques. This review article offers a detailed analysis of the current literature, serving as a valuable resource for researchers and professionals involved in developing small object detection and tracking systems.

### 2.2. YOLO v8

The YOLO v8 architecture consists of three main components: a backbone, a neck, and a decoupled head. The backbone employs an EfficientRep block, a convolutional neural network (CNN) designed to extract features from the input image. The neck, called the C2f module (C2f module refers to an optimized version of the CSP—cross stage partial bottleneck—with two convolutions, known as the C2 module), integrates these features from various depths of the backbone to generate a more detailed image representation. Finally, the decoupled head processes this fused feature map to estimate the size of the boxes surrounding the object in the image and the probability of the object being classified as a specific type [24], see Figure 1.

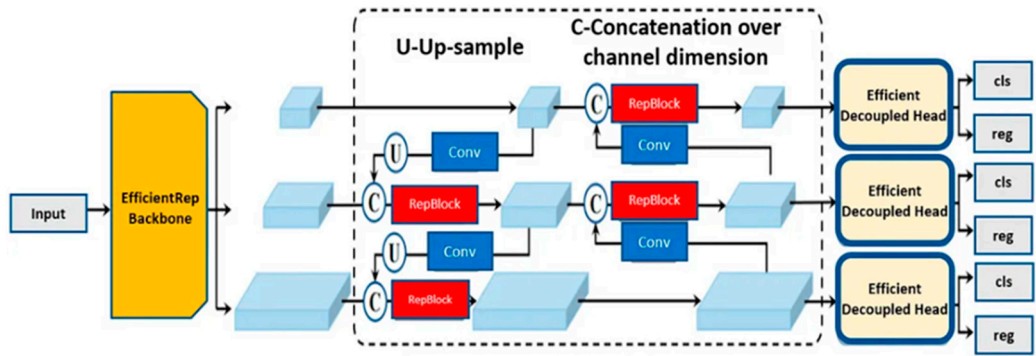

**Figure 1.** YOLO v8 architecture, adapted from [24].

The backbone is a lightweight CNN optimized for efficiency and accuracy. It comprises multiple convolutional layers, each designed to extract different levels of abstraction from the input image. Although the specific details of these convolutional layers are not depicted in the provided image, they are extensively documented in the YOLO v8 literature [24]. The neck, or C2f module, is a novel architectural feature introduced in YOLO v8. It aggregates feature maps from different depths of the backbone using a channel concatenation operation. This concatenation enables the network to learn relationships between features at various levels of abstraction, thereby enhancing the model's accuracy.

The decoupled head is tasked with generating the final predictions. It utilizes the fused feature map from the neck to predict bounding boxes and class probabilities for the objects in the image. The decoupled head includes several convolutional layers, culminating in one layer that predicts bounding boxes and another that predicts class probabilities.

Overall, the YOLO v8 architecture represents an advanced and efficient design for object detection. The EfficientRep backbone achieves an optimal balance between accuracy and speed, the C2f module facilitates the learning of richer feature representations, and the decoupled head ensures precise predictions of objects within the image. Research on aromatic herb identification has evolved significantly in recent years, addressing different approaches and methodologies. In this section, we review the main studies that form the

basis of our work, highlighting the most relevant contributions and the gaps that persist in the literature.

### 2.3. Related Works

Research on aromatic herb identification has evolved significantly in recent years, addressing different approaches and methodologies. In this section, we discuss the primary investigations that form the basis of our work, highlighting the most relevant contributions and the gaps that persist in the literature.

Che Soh et al. [25] investigated the process of herb identification, which is accomplished through organoleptic methods and relies heavily on botanical compounds; this makes it difficult to differentiate different species of herbs based on their aroma alone because of their shared physical attributes and aroma. Artificial technology is different from humans; unlike humans, it is thought to have the capacity to differentiate different species precisely. The authors developed an electronic nose that intended to recognize the scent of 12 species, including the Myrtaceae, Lauraceae, and Zingiberaceae families. The output received by electronic sensors for nasal gas was categorized using two different artificial intelligence techniques: ANN and ANFIS. From the outcome, ANFIS has a percentage of 94.8 compared to ANN's percentage of 91.7.

Previous research [26] studied the effectiveness of deep machine learning algorithms in classifying medicinal plant leaves from images. The investigation sought to create effective and accurate methods for differentiating between species of plants and diseases, using advanced technologies such as convolutional neural networks (CNNs) to improve the accuracy and reliability of plant recognition systems. The methodology included data collection and pre-processing steps, feature extraction, model development and training, and performance evaluation. The results indicated that using CNNs increased classification accuracy and efficiency, with high recognition rates and potential applications in environmental and agricultural monitoring.

Fernandes et al. [27] proposed a different approach, which was focused on creating an architecture that recognized 18 aromatic herbs using augmented reality and computer vision. The project and development of this architecture facilitated the overlay of virtual information onto the naturally identified environment, which contained an intuitive and simple-to-use mobile application. However, the primary obstacle was the creation of large databases for agricultural solutions with diverse and high-quality images. These images augmented the real world and provided solutions based on AI and neural networks for future endeavors.

Duth et al. [28] developed a classification system for medicinal herb leaves using digital image processing and machine learning techniques. The technologies associated with convolutional neural networks (CNNs), such as recurrent neural networks (RNNs), long short-term memory (LSTM), and object detection models like the region-based convolutional neural network (RCNN), Fast RCNN, and Faster RCNN, were employed. These technologies were employed to extract relevant features from leaf images and improve classification accuracy and speed. The proposed system achieved high accuracy and robustness in identifying different species of medicinal herb leaves. Compared to other existing techniques, the proposed method demonstrated effectiveness and accuracy in identifying the leaves of selected medicinal herbs.

In summary, although these studies provide a solid foundation for our research, there is a clear need for technologies to identify aromatic herbs. Our work aimed to develop and present a procedure for identifying three types of aromatic herbs using object detection algorithms through YOLO v8 architecture.

Training a network specifically for aromatic or medicinal herbs is crucial due to their visual similarity, which makes accurate identification challenging. Such a model enhances agricultural practices by improving crop management and harvesting, automating identification processes to reduce labor and increase efficiency, and enabling data-driven decision-making for better sustainability. Additionally, it ensures product authenticity for

consumers, fosters trust and satisfaction, and supports scientific research on plant characteristics and genetic diversity, leading to potential discoveries. These benefits highlight the importance of specialized deep learning models in agricultural and market operations.

### 3. Methods

The YOLO v8 model was employed for the real-time detection of three types of aromatic herbs (rosemary, mint, and bay), using a convolutional neural network architecture to process entire images and apply non-maximum suppression to refine predictions. Through training with a diverse set of annotated images, the model achieved mAP50 and precision measurements above 0.7, although it demonstrated recall values above 0.5, suggesting a trend towards over-detection. It was decided not to incorporate improvements to the YOLO v8 model, maintaining the original primary structure without changes. The decision was to assess the performance of the unmodified standard model in accurately identifying and categorizing aromatic herbs, solely utilizing the training dataset without any further adjustments to its architecture or hyperparameters.

The YOLO v8 model, with its advanced architecture and real-time processing capabilities, proved effective in identifying aromatic herbs. By following the CRISP-DM methodology, the study ensured a systematic approach to model development, from understanding the business requirements to deploying a robust solution in a practical setting.

The investigation intended to create a model for identifying some herbs commercialized at the largest Brazilian center for food distribution, the Company of Warehouses and General Warehouses of São Paulo (CEAGESP). Considering the proposed research problem, the study followed the Cross Industry Standard Process for Data Mining (CRISP-DM) model [29]. Table 1 presents the research stages.

**Table 1.** Summary of procedures performed using the CRISP-DM model.

| CRISP-DM Stage | Activity | Tool/Method |
|---|---|---|
| Business Understanding | Identify experts within the organization. Gather and outline needs and expectations. Verify the existence of image databases within the organization. | Visits and meetings with representatives from CEAGESP. |
| Data Understanding | Understand the available data. Evaluate the quality of the available data. Verify if the volume of data meets business needs. | Analysis of the formats and characteristics of the three selected herbs. |
| Data Preparation | Select data for analysis. Cleanse the data. Format the data appropriately. Select the model training environment. Choose the tool for creating bounding boxes. Understand the model metrics. | Creation of the database using photos (in JPG format) obtained from different cell phone models, NIKON cameras, Samsung tablets, and websites. Use of Google Cloud via Colab Pro and Colab Pro+ products. Utilization of the CVAT [30]. |
| Modeling | Select the appropriate technique for modeling. | Utilization of the YOLO V8 framework |
| Evaluation | Analyze performance metrics. | Analysis of the model's recognition accuracy. |
| Deployment | Prepare the analysis report. | Analyze the results. |

The selected herbs for training were rosemary, mint, and bay leaf. According to the botanical description [31–33], rosemary can have opposite leaves, lack petioles, and be simple, linear, and leathery, with stellate hairs on the underside that are whitish in color and dark green on the upper surface. Mint leaves are described as having opposite leaves, provided with short petioles, oblong to oval in shape, with toothed margins. Bay leaf has persistent, petiolate, alternate leaves, elliptical or lanceolate, leathery with undulated, entire smooth edges, bright green on the upper surface, and pale green on the lower surface.

Table 2 shows the leaves of the aromatic herbs used in the study. Following this instruction, bounding boxes were created on the images used for training.

**Table 2.** Reference of the trained aromatic herbs.

| Aromatic Herb | Leaf Reference |
|---|---|
| Rosemary | |
| Mint | |
| Bay Leaf | |

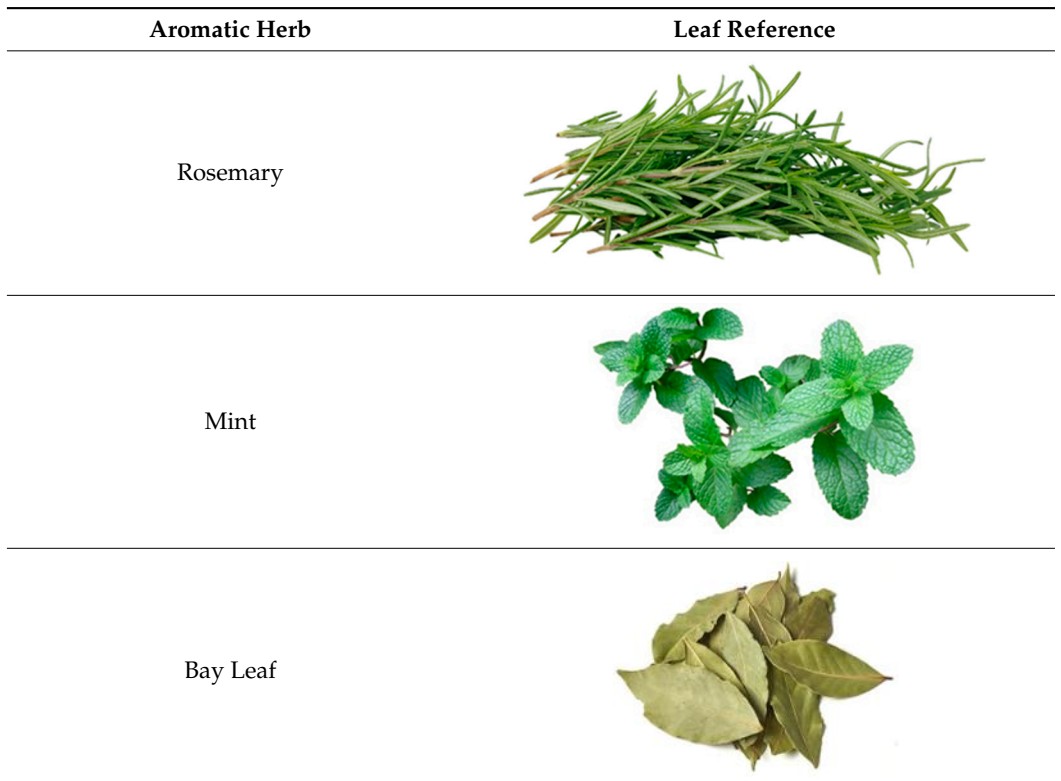

After consolidating the images, a cleaning process was conducted where images with significant blur were discarded. Finally, for each database, 10% of the images were selected for the validation process and 90% of the images were selected for the training process, resulting in the following:

- Rosemary leaf: 27 images for validation and 234 images for training.
- Mint leaf: 35 images for validation and 302 images for training.
- Bay leaf: 31 images for validation and 262 images for training.

The storage of the image database on the Google Cloud provider was utilized due to its connection with the Collaboratory platform [30], which is also from Google. Individual databases and a single database containing all three aromatic herbs were created for each herb. The classes for the single database containing the three aromatic herbs were classified as bay leaf (0), rosemary (1), and mint (2). The bounding boxes were created using the computer vision annotation tool [30].

The selection between the validation database and the test database was made randomly. In summary, the image database was analyzed and, taking care to prevent very similar images from being trained and validated, to avoid overfitting, the images were manually separated.

The images obtained during a visit to CEAGESP are herbs that have suppliers from different cities in São Paulo, Brazil. Regarding the maturity of the herbs, there was no specific selection concerning this requirement, as the focus has always been on identifying the herb regardless of its maturity. Consumers will face this condition daily; therefore, the database was not limited to maturity. The challenges regarding the identification of mint start from the definition of the bounding box; that is, as this herb has small leaves, it is necessary to define it by leaf or by the set of leaves, which are characteristics of the herb. When benchmarking with models available in Roboflow, there are these two situations.

The bay leaf is used fresh or dehydrated; therefore, like the other herbs in the study, they were not separated by maturity levels.

Furthermore, it has a variety of shapes as it originates from several tree species, with Loureiro and Umbellularia being the most common. Rosemary is a small evergreen shrub with thick aromatic leaves and many varieties. Regarding the identification challenges, the bounding boxes for each shrub are created.

The adopted performance metrics by class highlight explanations which are critical for understanding the model's performance for each class, especially in datasets with multiple object categories. For each class in the dataset, the following were calculated. (1) Precision (P): the accuracy of detected objects, indicating how many detections were correct (Equation (1)); (2) recall (R): the model's ability to identify all instances of objects in the images (Equation (2)); (3) mAP50: average precision (AP) is calculated at an intersection over union (IoU) threshold of 0.50. This measures the model's accuracy by considering only "easy" detections (Equation (3)); and (4) mAP50–95: the average of the average precision calculated at various IoU thresholds, ranging from 0.50 to 0.95. This provides a comprehensive view of the model's performance at different detection difficulty levels.

$$\text{Precision} = TP/(TP + FP) \tag{1}$$

where TP (true positives) = number of correct positive predictions, and FP (false positives) = number of incorrect positive predictions made by the model.

$$\text{Recall} = TP/(TP + FN) \tag{2}$$

where TP (true positives = number of correct positive predictions, and FN (false negatives) is the number of positives the model incorrectly predicted as negative.

$$mAP = \frac{1}{N} \sum_{i=1}^{N} AP_i \tag{3}$$

where $N$ = the number of classes and $AP$ = average precision.

As for the generated graphical measures, the following were adopted: (1) the F1 score curve represents the F1 score at various thresholds. Interpreting this curve can provide insights into the model's balance between false negatives and false positives at different thresholds; (2) the precision–recall curve, which is a graphical representation of the precision values at different thresholds; and (3) the recall curve, which is a graph that illustrates how recall values change at different thresholds.

In the training quantity, epochs are defined, meaning one epoch is a complete iteration of the entire training dataset in one cycle to train the model. During the epoch process, the model processes each training sample in the dataset, and its weights and biases are updated according to the calculated loss or error. The number of epochs is a hyperparameter that defines how often the learning algorithm will run through the entire training dataset. The training database is privately owned, and the programming logic for using YOLO v8 followed the instructions in the model document.

The model detects herbs individually, not by mixing them in a different group. This means that each herb is identified and analyzed on its own merits rather than being combined with other herbs to form a composite group. This approach allows for the more precise identification and analysis of each herb's unique characteristics and properties.

Training a deep learning model for identifying aromatic herbs substantially benefits from a comprehensive and high-quality training database. Such a database, comprising annotated images, enhances the model's accuracy and precision, reducing over-detection issues by providing more representative samples. It ensures the model's robustness against variations in herb appearance due to environmental factors and different cultivation practices. Additionally, it improves efficiency in agricultural operations by ensuring correct herb identification, thus streamlining processes in wholesale markets and retail.

## 4. Results and Discussion

The input of a neural network has weights, and these numbers are initialized randomly. In the learning process, these numbers are updated every time the neural network is stimulated, meaning the weights are adjusted towards the ideal value. There is no definition of a standard number of epochs for a neural network, but the present research aimed for a mAP50 greater than or equal to 0.7. In this scenario, a separate model was generated for each aromatic herb, considering 300 epochs. After this process, a single model was generated for all three herbs, also considering 300 epochs. Table 3 presents the performance metrics for each trained aromatic herb.

**Table 3.** Summary of training for the aromatic herb model.

| Aromatic Herb | Epoch Quantity | Time | mAP50 | mAP50–95 | P | R |
|---|---|---|---|---|---|---|
| Rosemary | 300 | 13 h | 0.739 | 0.369 | 0.724 | 0.786 |
| Mint | 300 | 15 h | 0.738 | 0.408 | 0.739 | 0.714 |
| Bay Leaf | 300 | 16 h | 0.7 | 0.387 | 0.611 | 0.727 |

The graphical measures of the F1 curve are shown in Figure 2a–c.

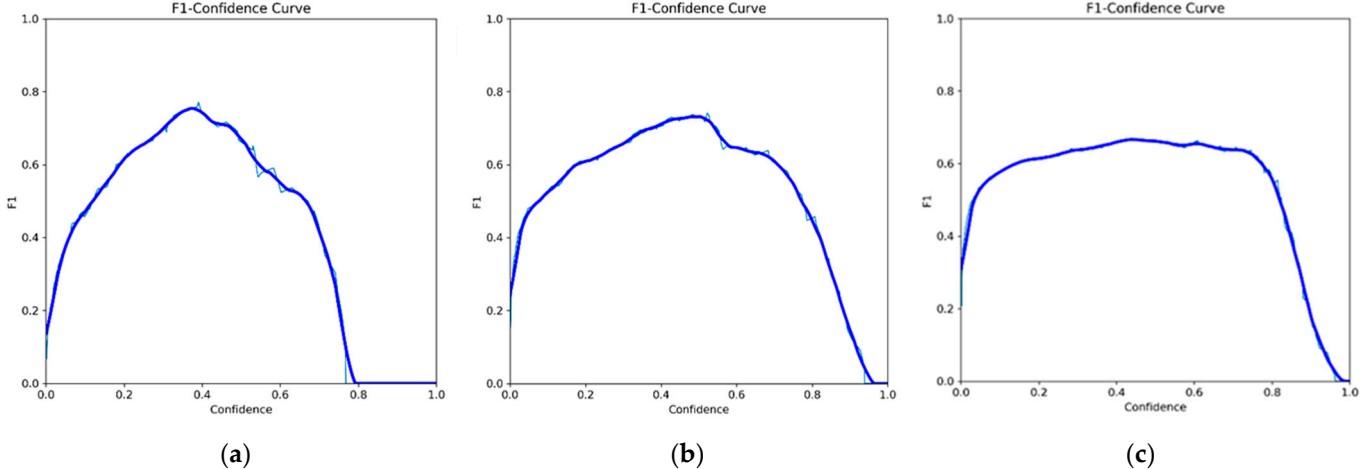

(**a**)  (**b**)  (**c**)

**Figure 2.** Graphical measures of the F1 curves for rosemary (**a**), mint (**b**), and bay leaf (**c**).

For the rosemary class, despite the mAP precision scores being above 0.7, it is noticeable that the recall measure exceeds the values of the previously mentioned measures, suggesting that negative objects are being incorrectly classified as positive. The F1 measure reinforces the need for improvement in detection, since it represents a balanced average of the model's ability to capture positive cases (recall) and to be precise with the cases it captures (precision). Additionally, it is noted that there is no need to change the number of epochs, since the mAP measure did not drop below 0.65 from 268 epochs onwards.

Although the recall measure for the mint class is below the precision and mAP measures, there is a slight difference. Therefore, it is understood that it follows the same behavior and improvements as the rosemary class. Additionally, the training occurred until 295 epochs, as there were no differences in the measures after this epoch.

Regarding the bay leaf class, the precision, mAP50, and F1 measures below the recall measure suggest that the object was not successfully identified with the trained sample. Concerning the variation in the mAP50 measure, after 192 epochs, the measure varied between greater than or equal to 0.6 and less than 0.7, meaning there were no measures above 0.7.

Google-enhanced computational resources (Colab Pro+) were used to train the model with the three aromatic herbs. Due to the machine's capacity and constant internet connection, the training occurred in six parts. Table 4 presents the performance metrics for the

trained consolidated model. According to Google, Colab Pro+ is a paid service offered by Google Colab, a cloud-based collaborative notebook platform. The central processing unit (CPU) is the general-purpose processing unit of a computer. While it is versatile and can handle a wide range of tasks, it may not be optimized for compute-intensive operations. On the other hand, the TPU (tensor processing unit) is custom-designed to accelerate machine learning workloads, particularly those that involve neural networks and large-scale data. Then, TPUs are highly specialized and can outperform GPUs in specific scenarios for training state-of-the-art deep learning models, especially when dealing with large datasets and complex neural networks in fields like natural language processing and computer vision.

**Table 4.** Summary of training for the consolidated model of aromatic herbs.

| Step | Epoch Quantity | Time (h) | mAP50 | mAP50–95 | P | R |
|---|---|---|---|---|---|---|
| 1 | 50 | 8 | All: 0.480<br>Bay Leaf: 0.479<br>Rosemary: 0.516<br>Mint: 0.445 | All: 0.221<br>Bay Leaf: 0.245<br>Rosemary: 0.203<br>Mint: 0.215 | All: 0.531<br>Bay Leaf: 0.561<br>Rosemary: 0.513<br>Mint: 0.52 | All: 0.507<br>Bay Leaf: 0.405<br>Rosemary: 0.750<br>Mint: 0.365 |
| 2 | 51 | 8 | All: 0.691<br>Bay Leaf: 0.595<br>Rosemary: 0.823<br>Mint: 0.653 | All: 0.321<br>Bay Leaf: 0.301<br>Rosemary: 0.328<br>Mint: 0.33 | All: 0.672<br>Bay Leaf: 0.552<br>Rosemary: 0.862<br>Mint: 0.602 | All: 0.724<br>Bay Leaf: 0.667<br>Rosemary: 0.821<br>Mint: 0.283 |
| 3 | 52 | 10 | - All: 0.715<br>- Bay Leaf: 0.612<br>- Rosemary: 0.862<br>- Mint: 0.671 | - All: 0.381<br>- Bay Leaf: 0.334<br>- Rosemary: 0.405<br>- Mint: 0.405 | - All: 0.692<br>- Bay Leaf: 0.617<br>- Rosemary: 0.802<br>- Mint: 0.658 | - All: 0.702<br>- Bay Leaf: 0.610<br>- Rosemary: 0.893<br>- Mint: 0.603 |
| 4 | 53 | 9 | - All: 0.761<br>- Bay Leaf: 0.664<br>- Rosemary: 0.942<br>- Mint: 0.679 | - All: 0.376<br>- Bay Leaf: 0.342<br>- Rosemary: 0.384<br>- Mint: 0.400 | - All: 0.778<br>- Bay Leaf: 0.618<br>- Rosemary: 1<br>- Mint: 0.715 | - All: 0.692<br>- Bay Leaf: 0.602<br>- Rosemary: 0.871<br>- Mint: 0.603 |
| 5 | 54 | 11 | - All: 0.737<br>- Bay Leaf: 0.642<br>- Rosemary: 0.871<br>- Mint: 0.698 | - All: 0.383<br>- Bay Leaf: 0.349<br>- Rosemary: 0.408<br>- Mint: 0.393 | - All: 0.743<br>- Bay Leaf: 0.655<br>- Rosemary: 0.857<br>- Mint: 0.718 | - All: 0.695<br>- Bay Leaf: 0.593<br>- Rosemary: 0.857<br>- Mint: 0.635 |
| 6 | 40 | 8 | - All: 0.723<br>- Bay Leaf: 0.619<br>- Rosemary: 0.884<br>- Mint: 0.666 | - All: 0.395<br>- Bay Leaf: 0.332<br>- Rosemary: 0.439<br>- Mint: 0.414 | - All: 0.754<br>- Bay Leaf: 0.661<br>- Rosemary: 0.919<br>- Mint: 0.682 | - All: 0.664<br>- Bay Leaf: 0.593<br>- Rosemary: 0.813<br>- Mint: 0.587 |

In the training of the three classes, it is noted that, concerning the measures presented, from the third training onwards, the variations in the measures were below 0.05. The recall and precision results showed the same characteristics as the individual training, i.e., values close together, indicating that the model is deficient in capturing positive cases (recall) and in being precise with the cases it captures (precision).

The class related to the aromatic herb rosemary presented mAP50, precision, and recall values above 0.8; thus, it was possible to recognize rosemary images during testing. However, when testing with an image of the herb chive, confusion in identification was noted, indicating an opportunity to add the chive class to the model, improve the database for the rosemary herb, and redo the bounding boxes, considering the herb, not the bunch.

The class related to the herb mint presented mAP50 and precision values close to 0.65, but the recall measure was also close to 0.6. The objective was to achieve a mAP50 value above 0.7. Considering the results of the other measures, one can understand the recognition confusion in the testing stage and the opportunities for improvement in the database for the mint herb and redo the bounding boxes. The class related to the bay leaf presented the worst performance measures. Despite the precision value of 0.66, the mAP50 and

recall values were close to 0.6. Therefore, there were various recognition confusions when considering different untrained aromatic herbs during the testing phase. Thus, there is an opportunity to improve the database for the bay leaf herb and redo the bounding boxes.

The challenges previously raised [8,18,34] were noted in the method stage, as image recognition became challenging due to variations in the aromatic herb's size, angle, and quality. These proportions or new or unusual configurations characteristic of aromatic herbs represent a challenge regarding the volume and diversity of the database to be used for training. Most difficulties we found in the present study, such as overlay detection and false positive detections, were also perceived by Weaver and Smith [32] when digitalizing an herbarium. We recommend increasing the number of images to overcome this issue.

After analyzing the results, we understand that the research's problem and objective have been achieved. Despite the opportunities for improvement in recognition through the improvement of the database and bounding boxes, it is noted that it is possible to train a deep learning model to identify diverse types of aromatic herbs.

Concerning the models' results for each herb and the model for the classes, there is an opportunity for improvement in the model training process for all aromatic herbs [17,18,35]. A model that does not produce false negatives, i.e., a model in which no undetected bounding boxes should be detected, has a recall of 1.0. Therefore, it is essential to classify negative samples. In the current approach, the mAP compares the ground truth bounding box with the detected bounding box. It returns a score, emphasizing the improvements to be implemented for a new training session.

Although this study was performed using the internet in real-time, there is an opportunity for training without internet dependency regarding the training environment when the training is conducted in an environment connected to the internet and from a cloud service provider. Thus, in addition to the cost of using this provider's environment, there were challenges regarding the constant connection of the environment. Internet speed does not influence training, but possible connection fluctuations do, because, as the training took place using resources from a public cloud, connection drops cause interruptions and continuity in training.

## 5. Conclusions

We found a model to automatically detect herbs (rosemary, mint, and bay leaf). The model with all classes showed mAP50 and precision measures greater than 0.7. However, it also showed values above 0.5 for the recall measure, indicating a possibility of over-detection. The recall measure indicates the ability of a model to find all the bounding boxes of the ground truth (positive samples); therefore, based on the results and discussions presented, the improvement of the database and bounding boxes stands out to better avoid noise in detection.

Concerning the database, as it was the first contact with the object under study and with the image identification process, the definition of the minimum number of images was presented in the study. Furthermore, no parameters were found to requalify the minimum quantity for training aromatic herbs in the literature. As mentioned, after analyzing the results, the need for database improvement is evident and is in progress.

**Author Contributions:** Conceptualization, S.N.A., W.A.C.L., J.C.L.F. and M.T.O.; methodology, S.N.A. and M.T.O.; software, S.N.A.; validation, W.A.C.L., J.C.L.F. and M.T.O.; formal analysis, M.E.F. and O.V.; investigation, S.N.A., W.A.C.L., J.C.L.F. and M.T.O.; resources, W.A.C.L., J.C.L.F., M.E.F., O.V. and M.T.O.; data curation, O.V.; writing—original draft preparation, S.N.A.; translation to English, F.P.L.A.; writing—review and editing, I.d.A.N.; visualization, M.T.O.; supervision, S.N.A.; project administration, M.T.O. All authors have read and agreed to the published version of the manuscript.

**Funding:** This research received no external funding.

**Data Availability Statement:** Data will be available from the corresponding author.

**Acknowledgments:** We appreciate the support from CEAGESP (Companhia de Entrepostos e Armazéns Gerais de São Paulo). This study was financed in part by the Coordenação de Aperfeiçoamento de Pessoal de Nível Superior—Brasil (CAPES)—Finance Code 001.

**Conflicts of Interest:** The authors declare no conflicts of interest.

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
