# Peer review of "Model Development for Identifying Aromatic Herbs Using Object Detection Algorithm"

_agriengineering, doi:10.3390/agriengineering6030112_

Round 1

Reviewer 1 Report

Comments and Suggestions for Authors

1. Line 36. The visual differentiation of herb leaves is challenging and it is one strong reason why we need to apply the YOLO approach. However, based on Table 2, the appearance of the three leaves (rosemary, mint, and bay leaf) is visually different. Please address more appropriate reasons.

2. Line 182. I think the authors should put more information on the samples used in the study. For example, the geographical origin (a GPS location for the plantation), and the maturity of the leaves. 

3. Line 196. How did the authors select the images for training and validation? Are there any prediction or testing samples to evaluate the model?

4. Line 308. Does the speed of the internet connection influence the result?

5. Line 313. Please show more explanation. What are the possible sources for the over-detection? 

Author Response

Reviewer 1

The authors thank the reviewer since the suggestions improved the manuscript considerably. Please find the answers just below the reviewer's comments.

  1. Line 36. The visual differentiation of herb leaves is challenging, and it is one strong reason why we need to apply the YOLO approach. However, based on Table 2, the appearance of the three leaves (rosemary, mint, and bay leaf) is visually different. Please address more appropriate reasons.

While the visual differences among rosemary, mint, and bay leaves are apparent, the YOLO approach's application is justified by its ability to handle variability in leaf presentation, speed and efficiency, robustness in complex environments, and high precision and recall. These features make YOLO a suitable and powerful tool for the automated identification of aromatic herbs in various practical applications. In lines 336 to 355, we detail the process of identifying herbs.

  1. Line 182. I think the authors should put more information on the samples used in the study. For example, the geographical origin (a GPS location for the plantation), and the maturity of the leaves.

The images obtained during a visit to CEAGESP are herbs that have suppliers from different cities in the state of São Paulo. Regarding the maturity of the herbs, there was no specific selection concerning this requirement, as the focus has always been on identifying the herb regardless of its maturity. Consumers will face this situation in everyday life; therefore, the database was not limited to maturity.

  1. Line 196. How did the authors select the images for training and validation? Are there any prediction or testing samples to evaluate the model?

The selection between the validation database and the test database was made randomly. In summary, the image database was analyzed and, taking care to prevent very similar images from being trained and validated, to avoid overfitting, the images were manually separated.

  1. Line 308. Does the speed of the internet connection influence the result?

Internet speed does not influence training, but possible connection fluctuations do because, as the training took place using resources from a public cloud, connection drops cause interruptions and continuity in training.

  1. Line 313. Please show more explanation. What are the possible sources for the over-detection?

The recall measure indicates the ability of a model to find all the bounding boxes of the ground truth (positive samples); therefore, based on the results and discussions presented, the improvement of the database and bounding boxes stands out to better avoid noise in detection.

Reviewer 2 Report

Comments and Suggestions for Authors

This paper develops and trains an image identification model for recognizing three types of aromatic herbs (rosemary, mint, and bay leaf). Even though this paper shows practical significance, it should be considered as an engineering application report rather than a research article. In addition, the following key issues and contents need to be reconsidered.

(1) The abstract needs to be rewritten, which did not mention any technical shortcomings.

(2) In the section Introduction, the proposal of the research question “Can a deep learning model be trained to identify diverse types of aromatic herbs correctly?” is not creative.  There are many deep-learning models proposed to recognize various herbs.

(3) The section Background needs to be rewritten. This section should describe the progress of related work on identifying aromatic herbs, rather than the explanation of existing technologies such as object detection models.

(4) The used dataset is too small, which is likely to cause overfitting.

(5) In the section Methods, there is no relevant content on model structure and method improvement.

(6) In general, this article does not propose targeted identification methods for the research objectives.

Comments on the Quality of English Language

 Minor editing of English language required.

Author Response

Reviewer 2

The authors thank the reviewer since the suggestions improved the manuscript considerably. Please find the answers just below the reviewer's comments.

This paper develops and trains an image identification model for recognizing three types of aromatic herbs (rosemary, mint, and bay leaf). Even though this paper shows practical significance, it should be considered as an engineering application report rather than a research article. In addition, the following key issues and contents need to be reconsidered.

1) The abstract needs to be rewritten, which did not mention any technical shortcomings.

The abstract was rewritten, mentioning the technical deficiencies.

2) In the section Introduction, the proposal of the research question "Can a deep learning model be trained to identify diverse types of aromatic herbs correctly?" is not creative. There are many deep-learning models proposed to recognize various herbs.

We changed the research question per the suggestion, which can be seen in lines 71 to 73.

3) The section Background needs to be rewritten. This section should describe the progress of related work on identifying aromatic herbs, rather than the explanation of existing technologies such as object detection models.

We completed the background session with a topic related to the progress of related work on identifying aromatic herbs.

4) The used dataset is too small, which is likely to cause overfitting.

In relation to the database, as it was the first contact with the object under study and with the image identification process, the definition of the minimum number of images was presented in the study. Furthermore, no parameters were found to requalify the minimum quantity for training aromatic herbs in the literature. As mentioned, after analyzing the results, the database improvement is evident and is in progress.

5) In the section Methods, there is no relevant content on model structure and method improvement.

We added model structure and method improvement to the methods section and added a topic related to the yoloV8 model in the background.

6) In general, this article does not propose targeted identification methods for the research objectives.

The article does propose targeted identification methods that are closely aligned with the research objectives. The use of the YOLO v8 framework, detailed data preparation and annotation processes, specific evaluation metrics, and a thorough discussion of results and practical applications collectively demonstrate the targeted nature of the methodology. The suggestions provided in the review will be carefully considered to refine further and enhance the clarity and impact of the research presented in the article.

Reviewer 3 Report

Comments and Suggestions for Authors

The authors propose to use computer vision systems to detect objects on herbal plants. The detector is based on the YOLOv8 neural network. The results are interesting, but there are a number of comments to the paper:

1) Line 21: YOLOv8 framework -> YOLOv8 model (Ultralytics is framework for YOLOv8)

2) In the review part, citations to sources on the application of early versions of YOLO in horticulture should be added (10.3390/electronics12030727, 10.3390/agronomy10071016).

3) Review papers (10.18287/2412-6179-CO-922, 10.3390/s23156887) are worth mentioning when considering the detection problem

4) Formula (3) is not represented by a formula but by a description in Latex-like style. Correction.

5) Need to add a figure with the YOLOv8 architecture in the Methods section

6) Need to specify the version (yolov8n,m,l?).

7) In Table 4, the column name indicates that the time is in hours. There is no need to duplicate this information in the cells.

8) Need to specify the GPU/TPU characteristics on the Colab Pro+ virtualized machine.

9) Anti-plagiarism report showed a level of 22% matches. It is desirable to add originality to the work.

Author Response

Reviewer 3

The authors thank the reviewer since the suggestions improved the manuscript considerably. Please find the answers just below the reviewer's comments.

The authors propose to use computer vision systems to detect objects on herbal plants. The detector is based on the YOLOv8 neural network. The results are interesting, but there are several comments to the paper:

1) Line 21: YOLOv8 framework -> YOLOv8 model (Ultralytics is framework for YOLOv8)

The line was corrected.

2) In the review part, citations to sources on the application of early versions of YOLO in horticulture should be added (10.3390/electronics12030727, 10.3390/agronomy10071016).

Articles were added in the background.

3) Review papers (10.18287/2412-6179-CO-922, 10.3390/s23156887) are worth mentioning when considering the detection problem

Articles were added in the background.

4) Formula (3) is not represented by a formula but by a description in Latex-like style. Correction.

The formula has been corrected.

5) Need to add a figure with the YOLOv8 architecture in the Methods section

Inserted as Figure 1

6) Need to specify the version (yolov8n, m, l?).

Yolo 8s.

7) In Table 4, the column name indicates that the time is in hours. There is no need to duplicate this information in the cells.

It was corrected as suggested.

8) Need to specify the GPU/TPU characteristics on the Colab Pro+ virtualized machine.

We included COLAB information on page 416.

9) Anti-plagiarism report showed a level of 22% matches. It is desirable to add originality to the work.

Through revisions and improvements to the text, we ensure that our work is original and contributes significantly to the field of study. We appreciate the suggestions and are confident that the changes have improved our manuscript's quality and originality, making it suitable for publication.

Reviewer 4 Report

Comments and Suggestions for Authors

The paper appears to be a master's thesis or course conclusion work that has been adapted to become an article. In this case, it must be redone as many terms in the text are not suitable for this form of publication, perhaps book chapters. The fact that there is little work on the identification of aromatic or medicinal herbs is not a reason "per se" that justifies work, but rather why there would be a need for a network to be trained for this purpose. Furthermore, due to the configuration being so far from the chosen herbs, networks trained for other purposes presented better solutions for greater challenges (extremely similar targets, for example), which makes the trained network very basic given the mAP50 values ​​obtained. It seems that they started the process and gave up, when an expansion of the training database would actually be enough to do so, as suggested in the discussion. The fact that they obtained a model that detects all herbs does not make it clear whether this detection occurred, for example, by mixing them in a group of herbs different from those trained, which in fact denotes that the methodology must be better described. Therefore, the suggestion is that the paper can be published, but it must undergo a substantial review.

Comments on the Quality of English Language

The English used allows the text to be read, but it is noted that it presents unusual phrases in native English, making reading difficult... it can be improved.

Author Response

Revisor 4

The paper appears to be a master's thesis or course conclusion work that has been adapted to become an article. In this case, it must be redone as many terms as possible in the text are not suitable for this form of publication, perhaps book chapters. The fact that there is little work on the identification of aromatic or medicinal herbs is not a reason "per se" that justifies work, but rather why there would be a need for a network to be trained for this purpose. Furthermore, due to the configuration being so far from the chosen herbs, networks trained for other purposes presented better solutions for greater challenges (extremely similar targets, for example), which makes the trained network very basic given the mAP50 values obtained. It seems that they started the process and gave up, when an expansion of the training database would be enough to do so, as suggested in the discussion. The fact that they obtained a model that detects all herbs does not make it clear whether this detection occurred, for example, by mixing them in a group of herbs different from those trained, which in fact denotes that the methodology must be better described. Therefore, the suggestion is that the paper can be published, but it must undergo a substantial review.

  • The article, titled "Development of a model for identifying aromatic herbs using object detection algorithm", is research in progress and we started with three aromatic herbs to train and evaluate the yolo v8 model and the final objective is 18 aromatic herbs, Therefore, some adjustments were made to make this work a compatible article for this magazine.
  • A review was performed to be more concise and focused and all terminology to be accurate and commonly accepted in the field of computer vision and deep learning.
  • The importance of identifying types of aromatic herbs through YoloV8 will improve agricultural practices, guarantee quality control in the distribution of herbs and assist in the automation of herb identification, as well as we detected that there is little research in this area.
  • The methodology was improved by including the yolo model, as well as detailing the research steps.
  • We have expanded the evaluation section to include a more detailed analysis of the results and why the current approach was chosen.
  • We explained in the text the chosen dataset and how we could improve the image enlargement. As we can see in table 4, we do not see major changes in the measurements between steps 5 and 6 and, therefore, the training was terminated, and the improvements (database and bounding box) were recommended for new training.
  • We make sure the results are clearly described and interpreted.

Round 2

Reviewer 1 Report

Comments and Suggestions for Authors

I agree with the revised version.

Author Response

The authors thank the reviewer since the suggestions improved the manuscript considerably.

Reviewer 2 Report

Comments and Suggestions for Authors

The authors have made extensive revisions to the comments provided, which basically meet the requirements. Nevertheless, some improvements are still needed before publication. (1) In the section Abstract, the authors are suggested to avoid abbreviations and strengthen innovative explanations. (2) In the section Background, the first three paragraphs are unnecessary.

Comments on the Quality of English Language

 Minor editing of English language required.

Author Response

Thank you for your valuable feedback. We have made extensive revisions based on the comments provided and believe we have addressed most of the requirements. However, we acknowledge that some improvements are still needed before publication.

  1. Abstract: We have revised the abstract to avoid abbreviations and strengthen the innovative aspects of our explanations.
  2. Background: We have reviewed the Background section and agree that the first three paragraphs are not essential to the core content of our paper. Therefore, we have removed these paragraphs to ensure a more concise and focused narrative.

Reviewer 3 Report

Comments and Suggestions for Authors

The authors have eliminated all comments. The work can be published.

Author Response

The authors thank the reviewer since the suggestions improved the manuscript considerably